

# Overtopping breaching of river levees constructed with cohesive sediments

H. Y. Wei[1], M. H. Yu[1], D. W. Wang[2], Y.T. Li[1]

[1]State Key Laboratory of Water Resources and Hydropower Engineering Science, Wuhan University, Wuhan, 430072,
People's Republic of China

[2]State Key Laboratory of Simulation and Regulation of River Basin Water Cycle, China Institute of Water Resources and
Hydropower Research, Beijing, 100038, People's Republic of China

*Correspondence to*: M. H. Yu (mhyu@whu.edu.cn)

**Abstract.** Experiments were conducted in a bend flume to study the overtopping breaching process and the corresponding
overflow rate of river levees constructed with cohesive sediments. The river and land regions were separated by the constructed
levee model in the bend flume. Results showed that the levee breaching process can be subdivided into a slope erosion stage, a
headcut retreat stage and a breach widening stage. Mechanism such as scour-hole erosion, flow shear erosion, impinging jet
erosion, side slope erosion and cantilever collapse were discovered in the breaching process. The erosion characteristics were
determined by both flow and soil properties. Finally, the levee breaching flow rates were simulated by a depth averaged 2-D
flow model. The calculated overflow rates can be well expressed by the broad-crested weir flow formula. The deduced
discharge coefficient was smaller than that of common broad-crested rectangular weirs because of the shape and roughness of
the breach.

## 1   Introduction

Levees, as one form of embankments, are constructed around rivers, lakes or seas to constrain flow and protect local residents
from flooding disasters (Schmocker and Hager, 2009). Levees constructed with cohesive sediment are the most common type
due to their low cost and the convenience with which construction materials can be acquired locally. When lacking a protection
layer, this kind of levee can be easily breached by overtopping flow if water level exceeds the design water level including the
freeboard (Pickert et al., 2011). Consequently, the protected area will be submerged, threatening lives and properties. Under
the circumstances, predicting flood propagation process and repairing the levee breach as soon as possible is crucial for
diminishing losses. For this, a profound understanding of cohesive levee breaching process and overflow rates is necessary.

Most studies performed on the overtopping breaching of embankments have focused on non-cohesive ones constructed normal
to river flow. Coleman et al. (2004) found the breach channel of overtopped embankments under constant water level has a



curved shape and the breach development obeyed the minimum energy dissipation rate rule for streams. Schmocker and Hager

(2009) studied the limitations on sediment size, dike width, dike height and unit discharge in laboratory experiments of non-cohesive dike breach. Pontillo et al. (2010) applied a one-dimensional two-phase model to simulate flow propagation and breaching process of a trapezoidal-shaped sediment dike, and the results were proved to be reasonable. Pickert et al. (2011) found that embankments composed of finer materials exhibited discontinuous erosion affected by cohesion due to pore-water pressure. Based on large amount of experiment data, Schmocker and Hager (2012) proposed some normalized formulae

including dimensionless equilibrium dike height, equilibrium dike volume and maximum breach discharge versus approach flow discharge, initial dike height and sediment diameter.

Compared to non-cohesive embankments, overtopping breaching of cohesive embankments is a more complex phenomenon involving impinging jet flows, vortex flow structures, soil erosion, soil mass failure and other similar problems (Zhu, 2006). In

spite of this, quite a few studies on the topic have been performed. Many researchers have found that headcut retreat is a predominant mode during the overtopping breaching process of cohesive embankments (Wahl, 2004) and many prediction models of headcut retreat rate were put forward (Hanson et al., 2001; Stein and LaTray, 2002; Zhao et al., 2013). Zhu (2006) developed a model for overtopping breaching of cohesive embankments in which the headcut erosion process was included. Hanson (1999), Zhu (2011) and the IMPACT project (Morris et al., 2005) all studied the cohesive embankment breach process

based on laboratory experiments and large-scale experiments. And the latter two were also concerned with the influence of soil properties, embankment geometry and so on. The IMPACT project (Morris et al., 2005) and Zhang et al. (2009) respectively conducted a series of dam break experiments and the latter also studied the cause of headcut erosion, double spiral flow at the dam crest and effect of soil cohesion on breach process.

There exist few researches on overtopping breaching of non-cohesive river levees. Kakinuma and Shimizu (2014) conducted large-scale experiments on breaching of river levees at the floodway of an actual river channel. The breach development was categorized into four stages according to breach progress and hydraulic characteristics. Yu et al. (2013) studied the influencing factors on the overtopping breaching process of non-cohesive levees and the breach discharge properties.

While studies on overtopping breaching of river levees constructed with cohesive sediments are scarce. The available study was conducted by Liang et al. (2002) on the numerical simulation of The Yellow River dike breach. The levee materials were a combination of non-cohesive and cohesive soils. The riverbank widening model proposed by Osman and Thorne (1998) was adopted in calculating the breach enlargement. Few experimental studies were available for this topic.   To obtain reliable data, a series of experiments were performed in a bend flume at Wuhan University. The dataset includes temporal breach profiles

and flow variations. The objectives of the work are:
   (1)  To study the general characteristics of levee breach process and the flow near the breach



(2) To investigate the effect of inflow discharge, soil porosity and soil water content on breach development

## 2 Experiment setup

Experiments were carried out in a bend flume with the plane shape of "U" and bed slope of about 1/1000. Fig. 1a shows the experiment layout. The levee was built from the bend apex to the exit of the flume and the cross-section of the levee is shown in Fig. 1d. Initially, flow was restrained by the levee and a dry area at the outer side of the levee was separated from the wet part. The dry area, which will be submerged after the levee breaching, is named as the land region. While the remaining wet reach is the river region. The initial breach was located at the place where overtopping breaching is most likely to appear according to characteristics of bend flow and its scale is shown in Fig. 1c. At the end of the flume, there were two sluice gates and the sluice gate at the river region was used to adjust the water level. Breach erosion was started when the water level surpassed the elevation of the initial breach.

Four automatic water-level gauges were placed at the point numbered from S1 to S4 shown in Fig. 1a. S2 and S4 were in the land region while the others in the river region. A topography meter was placed above the initial breach, measuring the breach profile variation with time by moving back and forth. Velocity changes near the breach were measured by an acoustic Doppler velocimeter (ADV). The location of velocity monitoring points ($M_1$ and $M_2$) is shown in Fig. 1b and Fig. 1d. The breach development was recorded by a camera fixed at the flume wall near the breach.

Experiment cases are listed in Table 1. Such main factors as upstream inflow discharge $Q$, soil porosity $e$ and water content $w$ of the levee materials were considered. The levee was constructed by the silt clay with grain size distribution shown in Fig. 2. As can be seen, the soil consists of 10% clay particles ($d<0.005$ mm), 70% silt particles (0.005 mm$<d<0.075$ mm) and 10% sand particles ($d>0.075$ mm). The liquid and plastic limit of the soil is 18.1% and 28.6% respectively, measured by liquid-plastic limit combined method. Other soil parameters such as density $\rho$, dry density $\rho_d$, soil cohesion $c$ and initial friction angle $\varphi$ are all affected by $e$ and $w$.

The levee was constructed by gradually adding soil and compacting, layer by layer. The soil density was confined to control the levee porosity. Before experiments began soil samples were selected from the levee to test $w$, $e$, $c$ and $\varphi$. The tested values are listed in Table 1. Flow entering the flume was adjusted until the discharge reached the desired value listed in Table 1. After that, the sluice gate at the river region was adjusted to ensure a very slow rise of the water level. The thin films were removed just before the flow overflowed the levee top. Then levee erosion process was started.



## 3 Levee breaching process

### 3.1 General description

The process of overtopping breaching can be subdivided into three stages according to the breach erosion characteristics shown in Fig. 3. The initial stage, characterized by flow shear erosion on the levee slope at the land side, is named as slope erosion

stage (shown in Fig. 3a). In this stage, there appeared some small scour holes on the land-side slope. Enlargement of the scour holes steepened the slope and then a large-scale scarp known as headcut developed (shown in Fig. 3b). This is the beginning of the second stage defined as headcut retreat stage shown in Fig. 3b, Fig. 3c, Fig. 3d and Fig. 3e. At the end of this stage, the breach cross-section was almost washed out and a deep gully formed (shown in Fig. 3f). Then flow in the gully began to erode the side slopes of the breach, which is the third-stage erosion defined as breach widening stage.

### 100 3.2 Slope erosion stage

Initial erosion of this stage usually occurred at the toe of the levee at the land side, due to large flow shear stress there shown in Fig. 4. Fig. 4 shows the water surface and flow shear stress distribution at an instant after overtopping of a levee calculated by Briaud et al. (2008) with a two-dimensional free surface flow model. The large negative shear stress at the toe is caused by flow separation due to sharp corners between the levee and bed surface (Briaud et al., 2008). If the shear stress there surpassed the

soil critical shear stress, cohesive soil blocks would be eroded and a scour hole (shown in Fig. 3a) appeared at the originally relatively smooth levee surface. The small scour holes increased bed roughness and flow turbulence, which in turn accelerated the local scour, making the scour hole enlarge. Due to the nonuniformity of the levee body, scour holes may also appear at other weak places but developed more slowly and finally merged into the large scour hole at the bottom.

### 3.3 Headcut retreat stage

According to the shape of the headcut appearing at this stage, the headcut can be categorized into two basic forms, single-step form (Fig. 5a and Fig. 5d) and multiple-step form (Fig. 5b and Fig. 5c). Initially, an incomplete single-step headcut appeared due to the bottom erosion (shown in Fig. 3b and Fig. 5a). After the headcut retreated to the brink of the levee crest, a multi-step headcut appeared due to the layer construction of the levee (shown in Fig. 3 c, Fig. 3d, Fig. 5b and Fig. 5c) and at last, all the steps disappeared and there appeared a complete single-step headcut (shown in Fig. 3e and Fig. 5d).


When the overflow velocity was small, the flow just streamed down along the vertical headcut surface where flow shear erosion (FSE) mainly occurred.





If the flow velocity increased, the overflow will be departed from the headcut surface, forming an impinging jet. For multi-step

headcut, when the flow velocity was not large enough, there may exist multiple small impinging jets due to the steps (shown in Fig. 3d and Fig. 5c). For single-step headcut or multiple-step headcut with large-velocity overflow, there existed a single impinging jet (shown in Fig. 3b, Fig. 3e, Fig. 5a, Fig. 5b and Fig. 5d). The single jet and small bottom jet directly impinged the non-erodible foundation, with part reflected toward the headcut as a reverse roller which undermined the headcut bottom. The other small upper jets, however, impinged the erodible platforms of the headcut steps, exerting both normal and shear stress on

the platforms. The platforms were eroded or collapsed and finally disappeared. The above erosion mode caused by impinging jets, was defined as jet impinging erosion (JIE).

It should be noted that in this stage FSE and JIE usually appeared in company with each other. The common combination style is that FSE happened above and the other happened below. Or they distributed alternatively along the river flow direction.

Except for the two erosion modes, discrete mass failure may also appear during the erosion process. The headcut retreated gradually by the above three soil destruction modes.

It can be discovered that the headcut height reduction, which was caused by flow shear erosion on the breach top, was much slower than the headcut backward migration. This made the overflow velocity quite small (about 0.2m/s), hardly influenced by

inflow discharge. So when soil properties were similar and the inflow discharge the same (Case 3 and Case 4, for example), headcut retreat rate was almost the same (shown in Fig. 6).

Furthermore, there exists a positive correlation between headcut retreat rate and soil water content while negative when it comes to soil porosity. For example, headcut of Case 2 with a larger soil porosity, retreat three times faster than that of Case 1

(shown in Fig. 6). And for Case 1 with higher soil water content, it costs much more time for headcut retreat process compared with Case 3 (shown in Fig. 6). A reasonable explanation for this phenomenon is that small soil porosity or large soil water content can result in large soil cohesion as shown in Table 1. While for cohesive soil, large cohesion between particles can enhance soil erosion resistance decisively.

### 3.4 Breach widening stage

During the breach widening stage, the side slopes of the breach under water were eroded by flow shear stress and retreated, rendering the remaining part above the water suspending as cantilever shown in Fig. 5f.

The breach widening process is shown in Fig. 7. It can be seen that initially, the migration rate of the upstream and downstream side slope of the breach was similar. But afterwards, the downstream side was eroded faster. This can be explained by the main

flow directions shown in Fig. 3f.





The migration rate of the side slopes was influenced by both the levee erosion resistance and nearby flow structures. In this period, when soil properties were the same, larger upstream discharge of Case 4 resulted in larger flow velocity near the breach and consequently a faster side slope retreat rate compared to Case 3. The same as headcut retreat rate, decrease of levee soil porosity and increase of soil water content can both accelerate breach widening.

The cantilever above the water surface can sustain itself for a while before collapse attributed to inner tensile stress of soil. Assuming that the cantilever is cuboid and sustains flexural deformation, then the forces acting on the cantilever in critical fracture state is shown in Fig. 8. In such state, the inner tensile stress on top of the fracture surface reaches the soil tensile strength $\sigma_t$ and the moment equilibrium equation of the cantilever per unit width is expressed as (Fukuoka, 1994)

$$\frac{1}{2}GL_c = \frac{1}{6}\sigma_t H_c^2 \tag{1}$$

where $G$ is the weight of the cantilever per unit width expressed as $G = \gamma L_c H_c$; $\gamma$ is the unit weight of the cantilever expressed as $\gamma = \rho g$. $L_c$ and $H_c$ are the length and height of the cantilever when fracturing, respectively.

The critical cantilever length can be deduced from Eq. (1) as:

$$L_c = \sqrt{\sigma_t H_c / 3\gamma} \tag{2}$$

Assuming that the soil tensile strength is 0.7 times of soil cohesion (Zhu, 2008), then the average critical cantilever length $L_c$ can be calculated by Eq. (2) and the results are shown in Table 2. It can be seen that the calculated $L_c$ rather approximates to the measured value, which proves the reasonability of the above assumption about the cantilever fracture.

## 4  Flow characteristics near the breach

### 4.1  Flow velocity and water level variation

The velocity monitoring point of Case 1 is located at $M_1$ while that of Case 2 is located at $M_2$ shown in Fig. 9. Fig. 10 shows variation with time of the water level at monitoring point S2 $Z$ (representing the flooding routine process of the land region) and the water surface velocity near the breach at monitoring point $M_1$ ($M_2$). Directions of the velocity are shown in Fig. 9. Direction $x'$ is horizontal and perpendicular to the levee axis pointing to the land region, direction $y'$ is horizontal and parallel to the levee axis pointing to the downstream while direction $z'$ is vertically upward. $Ux'$、$Uy'$ and $Uz'$ are respectively velocity in the $x'$, $y'$ and $z'$ direction. $U$ is the resultant velocity.



It can be seen that the variation of water level in the land region corresponded well with the breach height $H$, increasing gradually initially but sharply later along with a sharp drop of breach height. Then it kept almost constant, although the breach went on widening.

The variation of the resultant flow velocity was also related to the breach height changes. For Case 1 (shown in Fig. 10a),
before 75 min, $U$ increased slowly with gradual decrease of the breach height. From the time of 75 min, the breach height $H$ began to drop sharply and correspondingly, $U$ began to raise sharply with 1 minute lag. At the time of 78 min, $H$ reduced to the minimum value of zero and the flow velocity $U$ reached the maximum value of 0.7 m/s. After that, along with the increasing water level in the land region, $U$ dropped gradually and stabilized itself at the value of about 0.5 m/s.

General flow direction and its variation trend can be deduced from the magnitude of the component flow velocities. Initially, $Uz'$ was almost zero, demonstrating the two-dimensional characteristics of the flow. $Ux'$ was almost the same with $Uy'$ initially but surpassed it afterwards, indicating that flow was toward the breach due to the decrease of the breach height. With the sharp decrease of the breach height, there appeared different variation trend of $Uy'$ for Case 1 and Case 2. For Case 1 (shown in Fig. 10a), $M_1$ is located at the upstream side of the breach and it is inevitable that there exist component velocity
toward downstream direction so $Uy'$ increased and kept at a relatively larger value. While for Case 2 (shown in Fig. 10b), $M_2$ is located in the middle of the breach, flow velocity there was directly toward the land region so $Uy'$ decreased to almost zero.

## 4.2 Overflow rates

It is rather hard to acquire the flow rates over the levee breach directly by measurements. So a depth-averaged 2-D flow model
was established to compute them. In the calculating process, the breach geometry was modified at each time step according to the above measured headcut retreat and breach widening process.

### 4.2.1 Numerical methods

The governing equations, written as vector form, are given as follows:

$$\frac{\partial U}{\partial t} + \frac{\partial F}{\partial x} + \frac{\partial G}{\partial y} = S \tag{3}$$

where $U$, $F$, $G$ and $S$ are expressed as

$$U = \begin{bmatrix} h \\ hu \\ hv \end{bmatrix}, \quad F = \begin{bmatrix} hu \\ hu^2 + gh^2/2 \\ huv \end{bmatrix}, \quad G = \begin{bmatrix} hv \\ huv \\ hv^2 + gh^2/2 \end{bmatrix}, \quad S = \begin{bmatrix} 0 \\ gh(S_{bx} - S_{fx}) \\ gh(S_{by} - S_{fy}) \end{bmatrix} \tag{4}$$


where $x$, $y$ are Cartesian coordinates shown in Fig. 11a; $h$ is water depth, $u$, $v$ are flow velocity in the $x$ and $y$ direction, respectively. $S_{bx}$, $S_{by}$, $S_{fx}$ and $S_{fy}$ are respectively bed slope and friction slope in the $x$ and $y$ directions. $S_{fx}$ and $S_{fy}$ are expressed by Manning equation as follows

$$S_{fx} = \frac{n^2 u \sqrt{u^2 + v^2}}{h^{4/3}}, \qquad S_{fy} = \frac{n^2 v \sqrt{u^2 + v^2}}{h^{4/3}} \tag{5}$$

where $n$ is Manning roughness coefficient.

Equation (3) was discretized by finite volume method in space. The calculating grids are shown in Fig. 11b. At any calculating cell $(i, j)$, the equation was discretized as

$$\frac{dU_{i,j}}{dt} = -\frac{1}{\Delta x}\left( F_{i+\frac{1}{2},j} - F_{i-\frac{1}{2},j} \right) - \frac{1}{\Delta y}\left( G_{i,j+\frac{1}{2}} - G_{i,j-\frac{1}{2}} \right) + S_{i,j} \tag{6}$$

where $\Delta x$ and $\Delta y$ are respectively grid size in the $x$ and $y$ direction. The numerical fluxes $F$ and $G$ were calculated by third-order Runge–Kutta TVD method based on Riemman solvers that can capture flow discontinuity (Dou et al., 2014).

In time, forward Euler Difference scheme was used and Eq. (6) was again discretized as

$$U_{i,j}^{n+1} = U_{i,j}^{n} - \frac{\Delta t}{\Delta x}\left( F_{i+\frac{1}{2},j} - F_{i-\frac{1}{2},j} \right) - \frac{\Delta t}{\Delta y}\left( G_{i,j+\frac{1}{2}} - G_{i,j-\frac{1}{2}} \right) + \Delta t S_{i,j} \tag{7}$$

As the schemes are based on explicit algorithms, time step should satisfy Courant-Friedrichs-Lewy (CFL) criteria (Toro, 1999):

$$N_{cfl} = \Delta t \times \min\left( \frac{|u| + \sqrt{gh}}{\Delta x}, \frac{|v| + \sqrt{gh}}{\Delta y} \right) \leq 1 \tag{8}$$

where $N_{cfl}$ is Courant number; $\Delta t$ is computational time step; $N_{cfl}$ was valued much smaller than unity because of large bed elevation variation involved in calculation of levee break flow. Here, it was proved that the value of 0.03 for $N_{cfl}$ can achieve reasonable result.

Rectangular grids with size set as 0.5 cm×0.5 cm were deployed. To avoid zigzag boundary caused by rectangular grids, the calculating area was extended to a large rectangle $ABCD$ shown in Fig. 11a. Elevations of the area outside the flume were all assigned a larger value than the maximum water level in the flume. A simplified balancing-point method (Zhou, 1988) was used to simulate boundary conditions. The method is described in detail as follows. In the x direction, for example, it is assumed that the node numbered $(i, j)$ is outside the flume wall while the node $i+1$ is inside with water depth $h_{i+1}$, flow velocity





$u_{i+1}$ and $v_{i+1}$. Then the variables at the node $(i, j)$ are assigned $h_{i+1}$, $-u_{i+1}$ and $v_{i+1}$ respectively. This is on the assumption that

boundary node is in the middle of the two nodes and the boundary is vertical to the $x$ direction at the point. When grid size is

sufficiently small, the method is viable, without changing the real boundary obviously.

During the propagation process of levee break flow on riverbed, there may exist temporary dry nodes without water. To handle

the problems, a minimum water depth of 0.001m was defined and initially all dry nodes were set as the minimum water depth.

At a given simulation time, if water depth at a node was less than the minimum water depth, then the node is regarded dry and

velocity was set as zero.

The flow rates at the inflow boundary $CE$ were set as that listed in Table 1 for each case, and the measured water level process

at the point S4 was set as outflow boundary condition. Before overflow calculation, a steady-state flow condition in the river

region was calculated by given inflow rate in Table 1 and corresponding water level at the point S3. The steady-state flow

condition was set as initial condition of overflow. The value of Manning roughness coefficient (0.02) was acquired by

comparing calculated and measured water level.

The model has been verified by Dou et al., (2014).


The flow parameters for calculating the overflow rates were defined at the initial land-side brink of the levee top. According to

the computed velocity and water depth at each grid between the breach, average velocity and water depth at the brink can be

acquired, by which the breaching overflow rates were calculated.

### 4.2.2   Calculated results and analysis

The calculated overflow rates $Q_b$ are shown in Fig. 12. Initially, the overflow rate was very small and increased gradually due

to the large and slowly-reducing breach height. Then it increased sharply, with a rapid decrease of breach height. Just after the

breach height decreased to zero, the overflow rate reached its maximum value and then decreased. Finally, the overflow rate

kept almost stable at a certain value. It can be also seen that the maximum overflow rate for Case 4, with a larger inflow

discharge (28.53 L/s) than the other three (14.64 L/s), is also larger.


With the measured maximum flow velocity and the corresponding breach width and water depth, the maximum overflow rate

of each case can also be estimated. For example, the maximum flow velocity of Case 1 and Case 4 is 0.668 m/s and 0.83 m/s

respectively, and the estimated maximum overflow rate is about 24 L/s and 29.9 L/s, the same magnitude with the values of

18.6 L/s and 24.3 L/s acquired by the numerical method.




The breach overflow rates can also be simulated by broad-crested weir flow formula, which can be expressed as (Hager and Schwalt, 1994)

$$Q_b = C_d \sqrt{2g} B h_0^{1.5} \qquad (9)$$

where $B$ is the levee breach width and $h_0$ is approaching energy head. $h_0 = h' + v_a^2/2g$, with $v_a$ the approaching velocity and

$h'$ the water head above the breach crest. $C_d$ is dimensionless discharge coefficient, the value of which depends on the breach shape. When the effect of approaching velocity can be omitted, Eq. (9) can be expressed as

$$Q_b = C_d \sqrt{2g} B h'^{1.5} \qquad (10)$$

For the simplicity of analysis, $Q_b$, $B$ and $h$ are nondimensionalized by gravity acceleration $g$ and the initial breach height $H_0$ to

$Q_{b*}$, $B_*$ and $h_*$ respectively (Coleman et al., 2004). $Q_{b*} = Q_b / g^{0.5} H^{2.5}$; $B_* = B / H$; $h_* = h / H$. Then Eq. (10) can be replaced by

$$Q_{b*} = \sqrt{2} C_d B_* h_*^{1.5} \qquad (11)$$

Depending on the simulated overflow rates and measured breach hydraulic parameters, the relation between $Q_{b*}$ and $B_* h_*^{1.5}$ is

shown in Fig. 13. With a correlation coefficient of 0.93, the fitting relation of $Q_{b*}$ and $B_* h_*^{1.5}$ can be expressed as

$$Q_{b*} = 0.34 B_* h_*^{1.5} \qquad (12)$$

From Eq. (11) and Eq. (12), $C_d$ is deduced as 0.24, smaller than the discharge coefficient of common rectangular weirs (Hager and Schwalt, 1994). This may result from both the initial trapezoid shape of the breach cross-section and the large surface

roughness appeared later at the land side (Pařílková et al., 2012).

At the later stage of the headcut retreat, the overflow rate of Case 4 is much larger than that of the other three cases with the same $B$ and $h'$. This may result from larger approaching velocity caused by larger inflow discharge.

## 5  Conclusions

Overtopping breaching of levees constructed with cohesive soils can be subdivided into three stages according to breach erosion characteristics. The initial stage was defined as slope erosion stage, characterized by flow shear erosion on land-side slope and small scour hole at bank toe. The next stage was headcut retreat stage, in which singe-step form and multi-step form headcut can be discovered. Both flow shear erosion and jet impinging erosion existed in this stage. Headcut migration rate was



hardly influenced by inflow discharge but had more to do with soil properties. Increase of soil content or decrease of soil

porosity can both decrease headcut migration rate. The final stage was breach widening stage, including erosion of side slopes under water and the cantilever collapse above water surface. The migration rate of side slopes was affected by both soil erosion resistance and nearby flow characteristics. The maximum cantilever length, however, was influenced by soil density, average cantilever height and soil tensile strength.

Both the water level and flow velocity variation near the breach corresponded well with the breach height changes. The magnitude of the component flow velocities near the breach can indicate general direction of the breaching flow. Along with the decrease of the breach height, the breaching flow began to be toward the breach. And near the breach, flow direction at upstream point was more toward the downstream while flow direction of downstream point was directly toward the land region.


The calculated overflow rates variation trend was the same with that of flow velocity near the breach and the overflow rate increased along with the inflow discharge. By substituting the calculated overflow rates and measured breach size into the broad-crested weir flow formula, the discharge coefficient was deduced, which was smaller than that of common rectangular broad-crested weirs.


*Acknowledgements*. The work was supported by Natural Science Foundation of China [grant 11502174], [grant 11272240] and the Fundamental Research Funds for the Central Universities of China [grant 2042015kf0047].

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

Table 1 Experiment cases and parameters

| Case | $Q$ (L/s) | $w$ (%) | $e$ (%) | $\rho$ (kg/m³) | $\rho_d$ (kg/m³) | $c$ (KPa) | $\varphi$ (°) |
|---|---|---|---|---|---|---|---|
| 1 | 14.64 | 21.2 | 41.2 | 1920 | 1590 | 22.22 | 25.79 |
| 2 | 14.64 | 21.07 | 42.88 | 1870 | 1540 | 21.85 | 25.13 |
| 3 | 14.64 | 19.8 | 40.8 | 1910 | 1600 | 20.94 | 25.77 |
| 4 | 28.53 | 19.9 | 40.8 | 1920 | 1600 | 21.07 | 26.4 |

Table 2 Calculated critical cantilever length

| Case | $\rho$ (kg/m³) | Average $H_C$ (cm) | $c$ (KPa) | $\sigma_t$ (KPa) | Calculated $L_C$ (cm) | Measured $L_C$ (cm) |
|---|---|---|---|---|---|---|
| 1 | 1920 | 4.7 | 22.22 | 15.554 | 11.4 | 11.1 |
| 2 | 1870 | 4.4 | 21.85 | 15.295 | 9.9 | 8.9 |
| 3 | 1910 | 4.4 | 20.94 | 14.658 | 9.4 | 7.8 |
| 4 | 1920 | 4.5 | 21.07 | 14.749 | 9.8 | 8.1 |

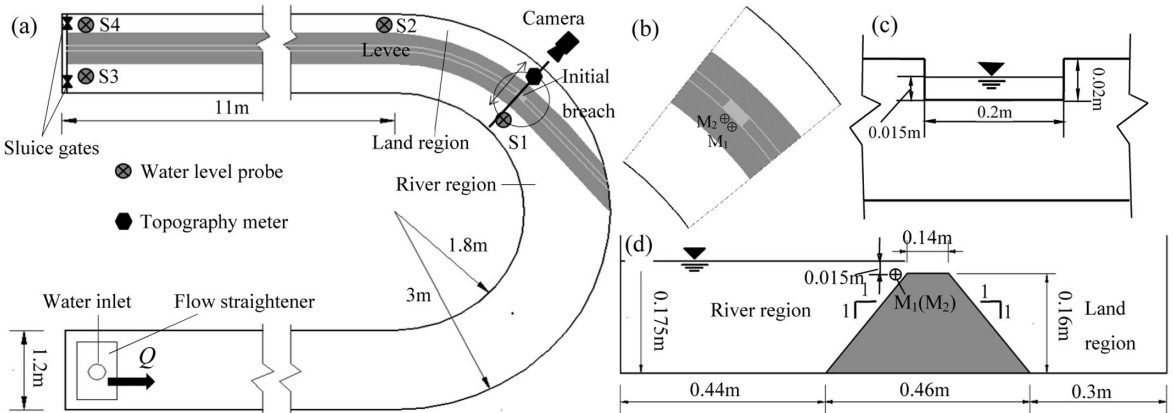

Figure 1 Levee model: (a) Top view of experiment layout, (b) Velocity monitoring point, (c) Longitudinal section of initial
       breach and (d) Transverse section of initial breach


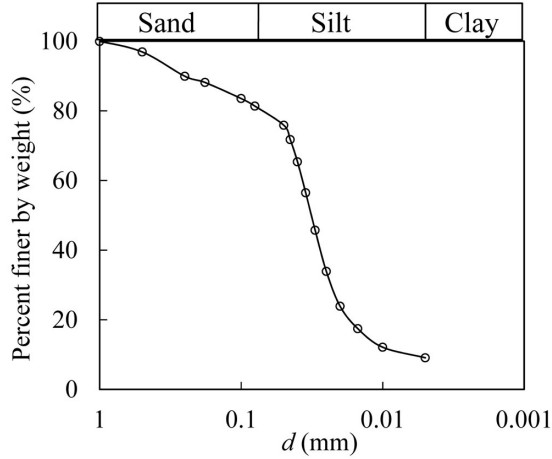

Figure 2 Grain size distribution of experiment materials


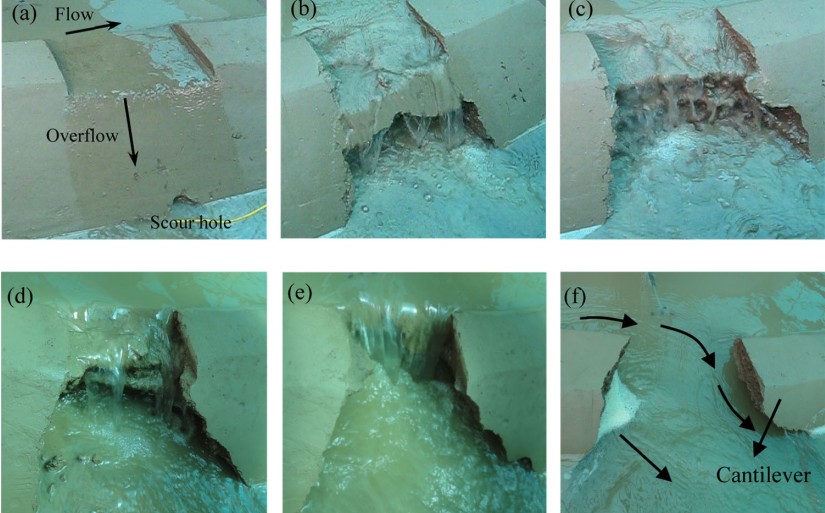

Figure 3 Levee breaching process: (a) Slope erosion stage; (b), (c), (d) and (e) Headcut retreat stage; (f) Breach widening stage

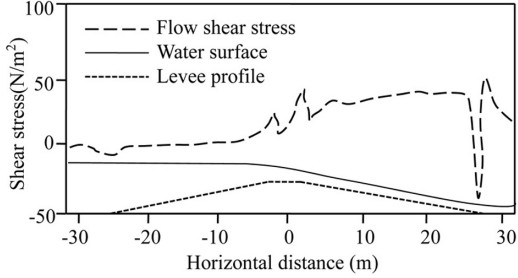

Figure 4 A typical flow shear stress distribution along the dike calculated by Briaud et al. (2008)




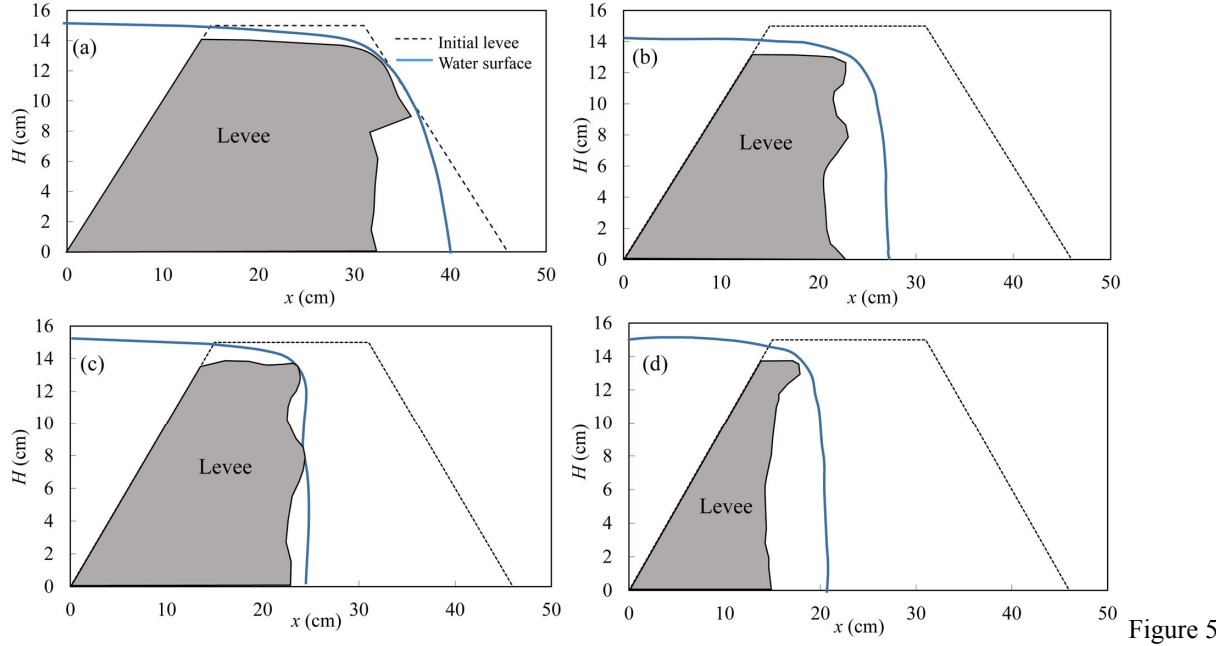

Figure 5

Headcut type: (a) Single headcut with single jet; (b) Multiple-step headcut with single jet; (c) Multiple-step headcut with multiple jets; (d) Single headcut with single jet


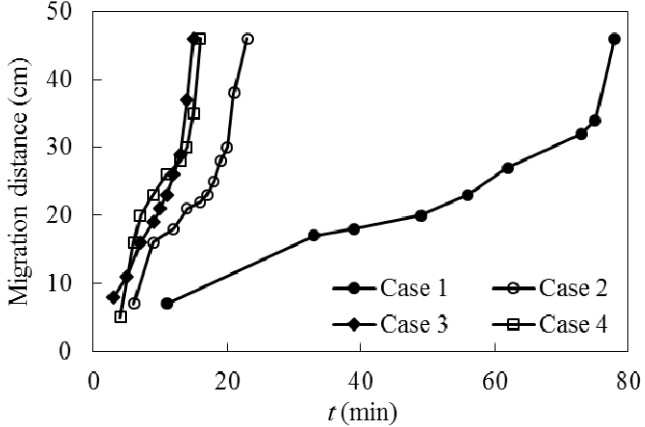

Figure 6 Headcut retreat process



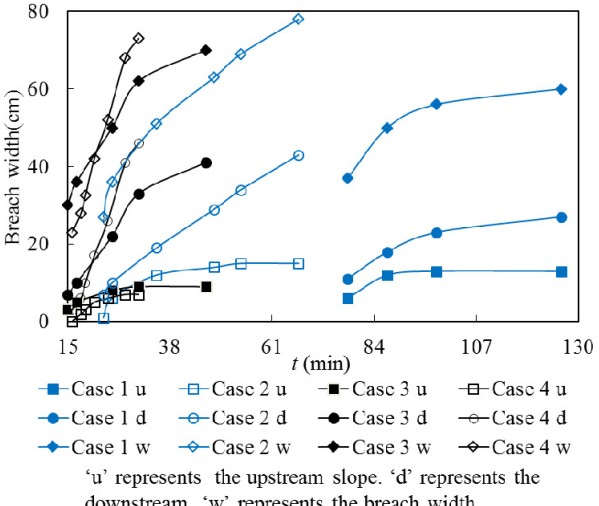

'u' represents the upstream slope. 'd' represents the downstream. 'w' represents the breach width

Figure 7 Breach widening process

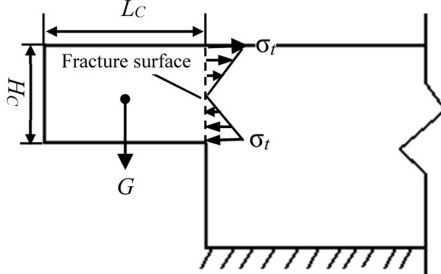

Figure 8 Critical fracture state of a cantilever

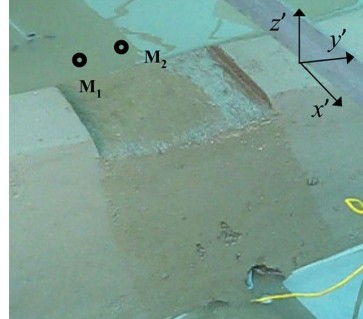

Figure 9 Velocity monitoring point


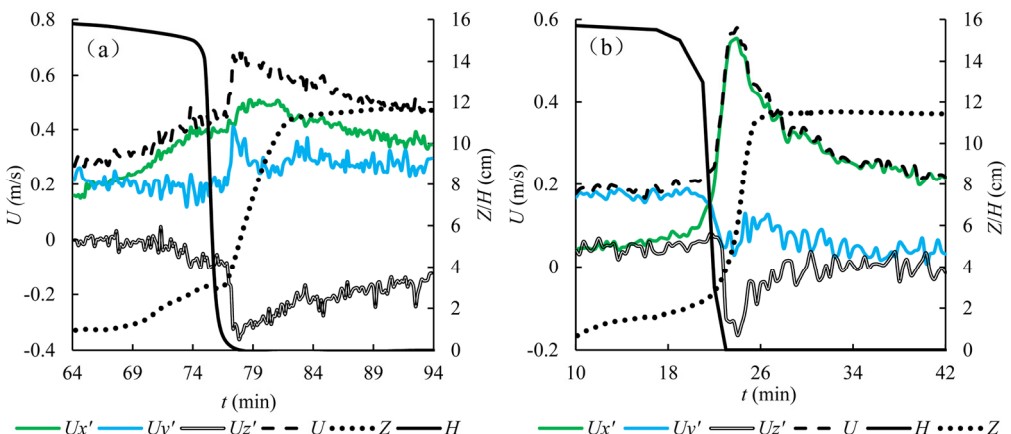

Figure 10 Relation between water level, velocity and breach height: (a) Case C-1, (b) Case C-2


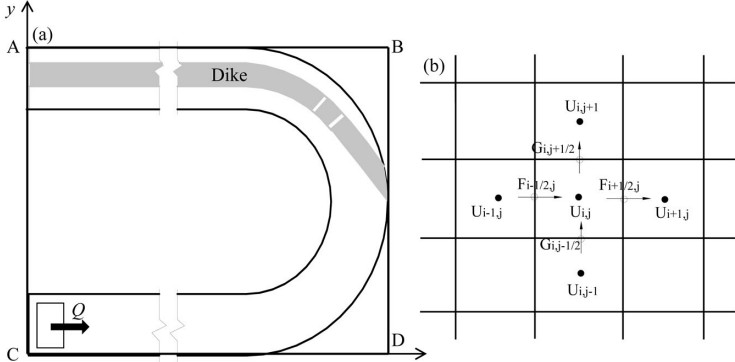

Figure 11 Calculated area

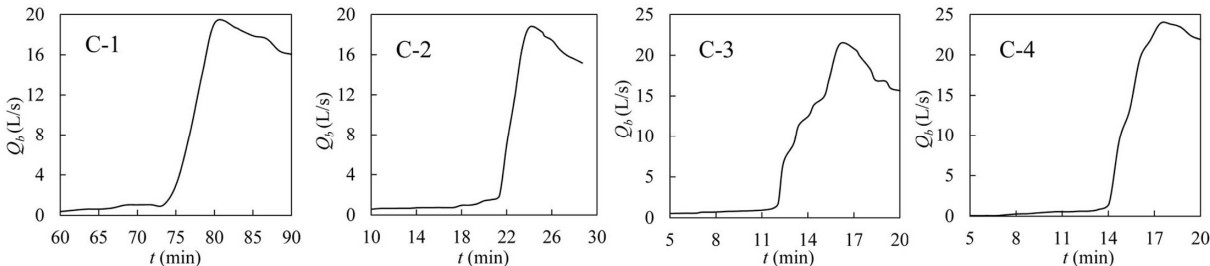

405                                 Figure 12 Simulated overflow rates





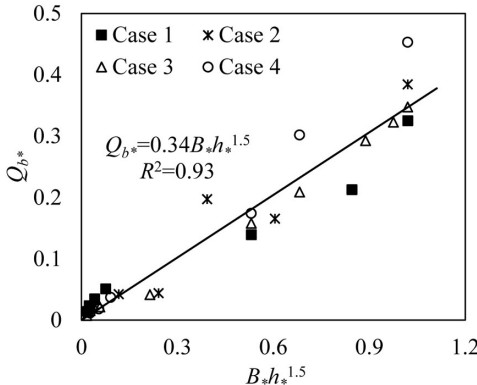

Figure 13 Fitting relation of $Q_{b*}$ and $B_*h_*^{1.5}$