# Peer review of "Overtopping breaching of river levees constructed with cohesive sediments"

_Natural Hazards and Earth System Sciences, 2015_

## Referee Comment (RC1) · G. Zhao (Referee) · 22 Feb 2016

General comments This article presents the experimental study of breach erosion in the cohesive levees and gives a simulation of the breaching flow with 2d numerical model. This study is very interesting and valuable to the get insights into the breach mechanism in cohesive levees and breaching flow. But what confused me is why the levee breach experiments were conducted in a bending channel flume. Has the breaching flow numerical model been validated before the applications to this study? The language should be generally improved. In General, the article can be acceptable before a major revision. Specific comments 1. In the introduction, there are no methodology and approach discussions in this study. Is it experiment in flume, numerical modelling? 2. Give a detailed explanation of the design of flume experiments. Why were they conducted in a bending channel flume? Why not do the tests in straight

channel flume? 3. ADV has been used in the velocity measurements in the flow with sediment. Has ADV been validated or calibrated since ADV has good accuracy in flow but not in the sediment-laden flow? 4. In Line 40, please give references to "Many researchers". 5. In Line 80, can you give a reference to the classification of "silt clay"? 6. In Line 83, can you give more detailed explanations of "other soil parameters"? or have you measured these values? 7. In Line 199, please give the reasons of choosing numerical model. In practice, the measurement should be chosen in the breaching test. 8. In line 199, you proposed to use a 2D flow model. Can you give a reference of this model? Has the model been validated to apply in the breaching flow? Does the model a non-hydrostatic module or a hydrostatic module? 9. In Line 281, eq. (12) has a different format with the traditional weir formula. Would you check eq. (12) from literatures? 10. It is better to have a discussion section before the conclusion section in Line 289 to discuss the experimental results and numerical modeling results with the past research. Technical corrections Line 47: Check the confusing sentence. Line 50: "There exist" should be "there are". Line 55: check the strange sentence. Line 58: To many space before "To obtain" Line 129: it is not suitable to use abbreviations of "FSE" and "JIE". And please check other abbreviations. Line 134: in "0.2m/s", there should be a space between value and unit. Line 249: "et al.," should be deleted ",". Check the sizes and formats of formulae in the text. They are not in the same format.

---

## Author Comment (AC1) · 10 Mar 2016

As there are some figures and tables in the reply, the reply to the comments has been uploaded in the form of a supplement.

Please also note the supplement to this comment:
http://www.nat-hazards-earth-syst-sci-discuss.net/nhess-2015-353/nhess-2015-353-AC1-supplement.zip

---

## Referee Comment (RC2) · M. Morris (Referee) · 8 Apr 2016

This paper presents an analysis of how breach occurs through cohesive materials and highlights many of the features that can be observed when levees and dams fail. Whilst demonstrating some of these processes, it also raises issues which are not considered in more detail - for example, how the erosion processes may vary as soil parameters vary; how scale effects may influence the processes and how flow around bends affects the breach formation process.

Working through the paper I noted the following points:

Abstract: Why perform these tests in a U shaped flume? If the focus of the work is to understand breach processes, it would be better to avoid the complication of curved approach flow conditions. If the intention was to analyse the effects of curved approach

flow, then a closer analysis of these aspects should be performed.

Ln 14 - why choose a 2D depth averaged model? Flow through breach is 3D and whilst it may be approximated to 2DV or 2DH for different aspects, any simplification will have an effect.

Ln 16 - choosing a fixed discharge coefficient for a weir shape that changes during the erosion process, and for which

Ln 19 - I don't understand the phrase " Levees, as one form of embankments" - what other forms are there which are not levees? I would choose one term (levee or embankment) and stay with it.

Ln 26 and numerous other references - you state a number of times that most studies look at non cohesives or cohesives. I would not agree. There are various researchers who have investigated both cohesive and non cohesive breach processes. In terms of large scale tests (ie. 1.5 - 2m high) tests, the most rigorous work on cohesive breach processes has been done by Hanson at HERU, USDA Stillwater. I'm not aware of similar scale and magnitude of tests on non cohesives, although there are plenty of smaller scale analyses. In conclusion, I would avoid suggesting any bias to one or the other.

Ln 38 - No - many of the processes listed also occur with non cohesives (eg. vortex flow, soil mass failure etc).

Ln 47 - you mean vortex rather than spiral flow?

Ln 89 - THIN FILMS - this is the only reference to thin films made in the paper. What is this referring to? It may be associated with trying to limit seepage through the test section before starting the test? If so, this is a key issue since the internal pore pressure conditions will affect how the test section performs (erodes) and this is subject to scale effects. How was this aspect dealt with in your tests?

Ln 138 - 144 - in practice you are referring here to the soil erodibility. How the erodibility

varies affects the headcut / erosion process. The soil state and parameters affect the erodibility and hence the breach process. A clearer analysis and understanding of these aspects is needed to place breaching processes in context for different levees, materials and material states. In your case, it should be recognised that the test results reflect a certain, specific soil type and state and hence some observed and calculated processes may vary as these parameters change.

Ln 148-150 - the approach flow conditions offer a whole area of research in relation to breach formation that could be / should be analysed!

Ln 168-170 - analysis of cantilever lengths. Whilst I agree with the observed process, and the approach to calculation is logical, there is considerable uncertainty in the 'rule of thumb' tensile strength value, as well as soil homogeneity - which means that the true strength in situ will vary unpredictably, so affecting the observed cantilever lengths.

Ln 261-265 - flow calculation: would have expected a closer fit between estimated and modelled values. maybe a closer look at the geometry of the flow control section would offer a means to reduce the uncertainty?

Ln 294 - what does "...increase of soil content..." mean? Suggest reword.

Fig 3a - you show a scour hole. Was there also seepage through the test material, and / or between the material and flume base?

Typos: Ln 12 - mechanisms Ln 24 - "...flood propagation processes..." Ln 119 - "...the overflow departs from the headcut surface..." Ln 146 - should read Fig 3f not 5f. Ln 292 - "...single step..."